# Capture of CO$_2$ Using Mixed Amines and Solvent Regeneration in a Lab-Scale Continuous Bubble-Column Scrubber

**Pao-Chi Chen \*, Jyun-Hong Jhuang, Ting-Wei Wu, Chen-Yu Yang, Kuo-Yu Wang and Chang-Ming Chen**

Department of Chemical and Materials Engineering, Lunghwa University of Science and Technology, Taoyuan City 33306, Taiwan; a0909562768@gmail.com (J.-H.J.); lokimumu0329@gmail.com (T.-W.W.); rockcat0921@gmail.com (C.-Y.Y.); roy29649@gmail.com (K.-Y.W.); gordon900429@yahoo.com.tw (C.-M.C.)
\* Correspondence: chenpc@mail2000.com.tw

**Abstract:** This study used monoethanolamine (MEA) as an amine-based solvent, which was blended with secondary amines (DIPA), tertiary amines, stereo amines, and piperazine (PZ) to prepare mixed amines at the required concentrations, which were used as the test solvents. To search for the best-mixed amines, a continuous bubble-column scrubber was adopted to explore the performance of mixed solvents presented in this study. The solvent regeneration test was also carried out at different temperatures. The selected factors included the type of mixed amine (A), the ratio of mixed amines (B), the liquid feed flow (C), the gas flow rate (D), the concentration of mixed amines (E), and the liquid temperature (F), each having five levels. Using the Taguchi experimental design, the conventional experimental number could be reduced from 15,625 to 25, saving much time and cost. The absorption efficiency (E$_F$), absorption rate (R$_A$), overall mass-transfer coefficient (K$_G$a), and absorption factor (φ) were estimated as the indicators. After the Taguchi analysis, E, D, and C were found to play important roles in the capture of CO$_2$ gas. Verifications of optimum conditions were found to be 100%, 19.96 × 10$^{-4}$ mole/s·L, 1.2312 1/s, and 0.6891 mol-CO$_2$/L·mol-solvent for E$_F$, R$_A$, K$_G$a, and φ, respectively. The evaluated indexes suggested that MEA + PZ was the best-mixed amine, followed by MEA and MEA + DIPA. The solvent regeneration tests for the scrubbed solutions performed at different optimum conditions showed that the heat of the regeneration sequence was in the order of MEA > MEA + PZ > MEA + DIPA with minimum energy required at 110 °C. The individual energy required was also analyzed here.

**Keywords:** scrubber; taguchi analysis; mixed amine; overall mass-transfer coefficient

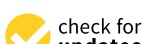



## 1. Introduction

Due to the influence of greenhouse gases, in which CO$_2$ is the major component, climate change is becoming gradually serious in the global world. Therefore, COP26 passed a resolution to control global warming below 1.5 °C before 2050 [1]. The emission amount of CO$_2$ was about 35 Gt in 2022, and the baseline emission is estimated to be 57 Gt in 2050. To keep the temperature rise below 1.5 °C before 2050, the emission of CO$_2$ needs to be maintained at 14 Gt; this means that a total amount of 43 Gt CO$_2$ needs to be reduced. According to the roadmap of the International Energy Agency, the reduction contribution in CO$_2$ emission using CCSU (Carbon Capture, Storage and Utilization) technology is at least 8 Gt. This shows that CCSU will be important in the coming 30–40 years. Recently, a Net-Zero-CO$_2$ emission between 2040 and 2060 was also addressed through three routes: carbon capture and storage (CCS), carbon capture, and utilization (CCU), and the use of biomass grown and processed for the specific purpose of making chemicals (BIO) [2]. This impact was largely for some CO$_2$ emissions sectors in the industry such as coal-fired plants, the petroleum industry, the cement industry, and the steel mill industry. To reduce CO$_2$ emission, several technologies are used, such as absorption, adsorption, membrane separation, freezing technology, and chemical looping [3], in which absorption was found

to be more effective in the $CO_2$ concentration range of 0–20% under 10 atm [4]. Recently, competing methods for the absorption and utilization of $CO_2$, such as the capture of $CO_2$ in molten salts, were reported in the literature [5,6]. The method used molten salt $CO_2$ capture and an electrochemical transformation process to obtain carbon at the cathode and oxygen at the anode. In addition, synthesizing oxygen reduction reaction catalysts from $CO_2$, connecting the fields of carbon capture and fuel cell research was reported in the literature [7].

Generally speaking, a double-unit process including a combination of scrubber and stripper is required when using the absorption method, in which the alkanolamine solutions were proposed as the absorption solvents for $CO_2$ capture before 2030 [8–11]. However, the capture cost in the stripping step in CCS is about 60–70%, making the selection of strippers and solvents pivotal. Due to this, several solvents have been adopted, such as amines [11–15], amino salts [16,17], ammonia water [18,19], sodium hydroxide [20], potassium carbonate [21], piperazines [22], ionic liquids, and physical solvents [10], with amines being the most popular solvents in these capture processes. The drawbacks of a single solvent, such as loading, corrosion rate, degradation rate, and regeneration energy, were improved by adopting new solvents, including mixed amines and non-aqueous solvents [23–28]. There are four kinds of amines, basically classified as primary amines (such as monoethanolamine, MEA) [29,30], secondary amines (such as diisopropanolamine, DIPA) [31–33], tertiary amines (such as triethanolamine, TEA) [34–36], and steric amine (such as 2-amino-2-methyl-1-propanol, AMP) [37–39]. Structurally, these amines have at least one hydroxyl group and one amino group, as shown in Figure 1. The presented hydroxyl group can reduce the vapor pressure, and the amino group can absorb acidic gases; therefore, it affects the physical properties of amines (Table 1). The vapor pressures at 20 °C for MEA, DIPA, and TEA are 64 Pa, 2 Pa, and 1 Pa, respectively. The vapor pressure of amine decreases with an increase in the –OH group. The loadings of various amines are affected by the functional groups. For instance, theoretically, the loading of MEA is limited stoichiometrically to 0.5 moles $CO_2$ per mole amine. However, the loading is higher for secondary and tertiary amines, with loadings of up to 1 mole of $CO_2$ per mole of amine. To promote the absorption rate, and decrease oxygen degradation and thermal degradation, aqueous piperazine [40], and a cyclic diamine shown in Figure 1e, solution was used to test for the capture of $CO_2$ [10,22].

**Figure 1.** Structures of various amines showing the difference in functional groups.

**Table 1.** Physical and chemical properties of amines [10,39–42].

| Items | MEA | DIPA | TEA | AMP | PZ |
|---|---|---|---|---|---|
| M.W. (g/mol) | 61.084 | 133.19 | 149.188 | 89.138 | 86.136 |
| Density (g/cm$^3$) | 1.0117 | 0.992 | 1.126 | 0.934 | 1.100 |
| BP (°C) | 170 | 249 | 335.4 | 165.5 | 146 |
| Solubility in water at 20 °C | Miscible | Miscible | Miscible | Miscible | Miscible |
| Vapor pressure (pa) (20 °C) | 64 | 2 | 1 | 40 | 10.66 |
| pK$_a$ | 9.50 | 8.80 | 7.76 | 9.70 | 9.78 |
| Reaction rate constant, $k_2$ (m$^3$/s·kmol) (25 °C) | 3630 | 2585 | 2202 | 810.4 | 48,533 |
| Activation energy $E_a$ (kJ/mol) | 41.2 | 39.9 | 36.9 | 41.7 | 33.7 |

The absorption process of $CO_2$ using an amine solution generally includes diffusion from the bulk gas phase to the gas–liquid interface, from interface diffusion into the liquid bulk phase, and reaction with an amine. The reaction between $CO_2$ and the amine can be described by a two-step zwitterions mechanism [43]. First, the reactions of $CO_2$ with a primary amine and secondary amine ($CO_2/H_2O/R_1R_2NH_2$) systems, based on the zwitterion system, are found to be [43,44]:

$$H_2O \rightleftharpoons H^+ + OH^-, \tag{1}$$

$$CO_2 + OH^- \rightleftharpoons HCO_3^- \tag{2}$$

$$CO_2 + H_2O \rightleftharpoons HCO_3^- + H^+, \tag{3}$$

$$CO_2 + R_1R_2NH \rightleftharpoons R_1R_2NH^+COO^- \tag{4}$$

$$R_1R_2NH^+COO^- + R_1R_2NH \overset{k_{amine}}{\rightarrow} R_1R_2NCOO^- + R_1R_2NH_2^+, \tag{5}$$

$$R_1R_2NH^+COO^- + H_2O \overset{k_{H_2O}}{\rightarrow} R_1R_2NCOO^- + H_3O^+, \tag{6}$$

$$R_1R_2NH^+COO^- + OH^- \overset{k_{OH^-}}{\rightarrow} R_1R_2NCOO^- + H_2O. \tag{7}$$

In Equations (2) and (3), the reaction constants are $k_{OH^-}$ and $k_{H_2O}$, respectively. In Equation (4), the forward reaction and reverse reaction constants are $k_2$ and $k_{-1}$, respectively. In Equations (5)–(7), the amine, $OH^-$ and $H_2O$ can be expressed as a base B, i.e., the three equations can be expressed as a single equation:

$$R_1R_2NH^+COO^- + B \overset{k_b}{\rightarrow} R_1R_2NCOO^- + BH^+. \tag{8}$$

The forward reaction rate equation at quasi-steady state becomes:

$$r_{CO_2,amine} = \frac{k_2[CO_2][R_1R_2NH]}{1 + \frac{k_{-1}}{\sum k_b[B]}}. \tag{9}$$

In most case, $k_{-1}/\sum k_b[B] << 1$ resulting simple second-order kinetics is obtained:

$$r_{CO_2,amine} = k_2[CO_2][R_1R_2NH]. \tag{10}$$

Considering the reactions (2) and (3), the overall rate was found to be:

$$r_o = r_{CO_2,amine} + r_{CO_2,H_2O} + r_{CO_2,OH^-} = k_{obs}[CO_2]. \tag{11}$$

The total rate of all $CO_2$ reactions in an aqueous solution is thus represented by the sum of the reaction rates. Where the $k_{obs}$ is:

$$k_{obs} = \left( \frac{k_2[R_1R_2NH]}{1 + \frac{k_{-1}}{\sum k_b[B]}} \right) + k_{H_2O}[H_2O] + k_{OH^-}[OH^-]. \tag{12}$$

If $k_{-1}/\sum k_b[B] << 1$, the $k_{obs}$ becomes

$$k_{obs} = k_2[R_1R_2NH] + k_{H_2O}[H_2O] + k_{OH^-}[OH^-]. \tag{13}$$

The tertiary alkanolamines (denoted here as $R_3N$ such as TEA) and PZ can also be expressed as shown in Equation (11) for similar kinetics as $[R_1R_2NH]$ with replacement of $[R_1R_2NH]$ by $[R_3N]$ or $[PZ]$ [38,39]. The reaction of $CO_2$ with $H_2O$ is usually neglected in the overall reaction rate equation resulting the terms of $r_{CO_2,H_2O}$ in Equation (11) and $k_{H_2O}[H_2O]$ in Equation (13) can be neglected.

The reaction of $CO_2$ with amines found that the reaction rate constant with $CO_2$ are 3630 [39], 2585 [39], and 2202 $m^3/s\cdot kmol$ [37] for MEA, DIPA, and TEA, respectively. However, the reaction constant is affected by pKa and temperature [40,45]. At a given temperature, $\ln k_2$ is proportional to $pK_a$. In addition, the activation energy in Table 1 shows that reactions with high $E_a$ is very temperature-sensitive and reactions low $E_a$ is very temperature-insensitive [46]. However, the reaction rate constants for AMP and PZ were 810.4 and 48,533, respectively. The reaction rate constant for PZ is much higher than the other amines. Therefore, the addition of PZ into amines has a powerful potential to promote the absorption rate. For mixed amines, the overall rate can be expressed as:

$$r_{o,Mixed} = \sum_i r_{CO_2,i} + r_{CO_2,OH^-} = k_{obs,mixed}[CO_2], \tag{14}$$

where $r_{CO_2,i}$ is the reaction rate of $i$th amine and $k_{obs, Mixed}$ is overall rate constant of mixed amines. Equation (14) states that the reaction rate is affected by concentration of various amine, $[OH^-]$ and $[CO_2]$. In addition, the gas–liquid contact method and temperature are also affecting the reaction rate [46].

Here, the means of $CO_2$ capture was explored to search for the best-mixed amine using a combination of amines including MEA, MEA + DIPA, MEA + TEA, MEA + AMP, and MEA + piperazine (PZ). In the capture process, however, several important factors need to be considered, such as the ratio of mixed amines, the liquid-flow rate, the gas-flow rate, the concentration of total amines, and the liquid temperature. In this study, the $CO_2$ absorption experiment was performed by using a continuous bubble-column scrubber because bubble columns have higher mass-transfer coefficients in the range of $0.04$–$1.54\,s^{-1}$ and higher specific surface area in the range of $100$–$1500\,m^2/m^3$ compared with packed bed, whose mass-transfer coefficient and specific surface area are in the range of $0.02$–$0.38\,s^{-1}$ and $100$–$600\,m^2/m^3$, respectively. The larger the mass-transfer coefficient, the smaller the column size [47,48]. In addition, the bubble column is comparable with packed bed although the pressure drop for a bubble column was higher than a packed bed. The scrubbing factor for the former is higher than the latter [20]. Due to this, the bubble-column has been found that it shows a superior performance, such as a high absorption, high mass-transfer coefficient, simple structure, higher scrubbing factor, and easy operation, compared with other scrubbers. In order to evaluate the performance of solvents, the indexes of solvents include the removal efficiency ($E_F$), the absorption rate ($R_A$), the overall mass-transfer coefficients ($K_Ga$), and the scrubbing factor ($\phi$).

To achieve this purpose, the Taguchi method was used in the experimental design to reduce the cost and save time. From the experimental data, the optimum conditions and sequence of parameters could be obtained through the S/N (signal/noise) ratio analysis. Verification of optimum conditions was required. Accompanying various indexes, the best-mixed amines for the $CO_2$ capture process could be obtained. Finally, solvent regeneration

for the best amine was also explored. A schematic of the research framework is shown in Figure 2.

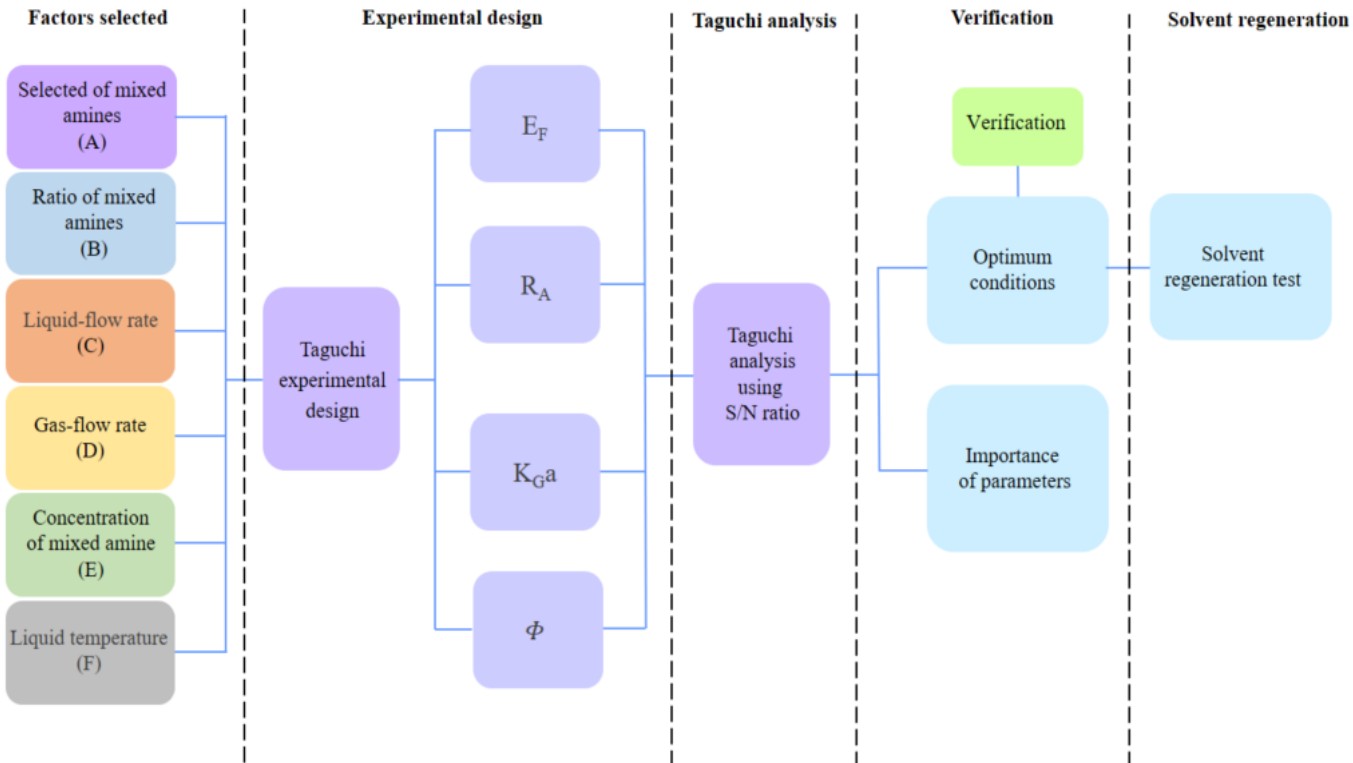

**Figure 2.** Research framework conducted in this study.

## 2. Methodology

### 2.1. Experimental Design

Six factors, including the type of mixed amines (A), the ratio of mixed amines (B), the liquid-flow rate (C), the gas-flow rate (D), the concentration of total amines (E), and liquid temperature (F), were assessed, each with five levels. According to traditional experimental design, a total of 15,625 runs had to be performed, leading to high costs and a longer time for experimentation. Therefore, using the Taguchi experimental design, the experimental number could be largely reduced to 25 runs, cutting the experimental cost up to 99.84%. The optimum conditions and sequence of importance could also be obtained from Taguchi analysis. Table 2 presents the factors and levels in this work, while Table 3 presents the orthogonal arrays with 25 runs and operating conditions.

**Table 2.** Factors and levels selected in this research.

| Factors/Levels | Level 1 | Level 2 | Level 3 | Level 4 | Level 5 |
|---|---|---|---|---|---|
| A (-) | MEA | MEA + DIPA | MEA + TEA | MEA + AMP | MEA + PZ |
| B (wt%) | 5 | 10 | 15 | 20 | 25 |
| C (mL/min) | 150 | 200 | 250 | 300 | 350 |
| D (L/min) | 4 | 6 | 8 | 10 | 12 |
| E (M) | 1 | 1.5 | 2 | 2.5 | 3 |
| F (°C) | 25 | 30 | 35 | 40 | 45 |

**Table 3.** Orthogonal arrays, $L_{25}(5^6)$.

| No. 1 | 1 | 1 | 1 | 1 | 1 | 1 |
|---|---|---|---|---|---|---|
| No. 2 | 1 | 2 | 2 | 2 | 2 | 2 |
| No. 3 | 1 | 3 | 3 | 3 | 3 | 3 |
| No. 4 | 1 | 4 | 4 | 4 | 4 | 4 |
| No. 5 | 1 | 5 | 5 | 5 | 5 | 5 |
| No. 6 | 2 | 1 | 2 | 3 | 4 | 5 |
| No. 7 | 2 | 2 | 3 | 4 | 5 | 1 |
| No. 8 | 2 | 3 | 4 | 5 | 1 | 2 |
| No. 9 | 2 | 4 | 5 | 1 | 2 | 3 |
| No. 10 | 2 | 5 | 1 | 2 | 3 | 4 |
| No. 11 | 3 | 1 | 3 | 5 | 2 | 4 |
| No. 12 | 3 | 2 | 4 | 1 | 3 | 5 |
| No. 13 | 3 | 3 | 5 | 2 | 4 | 1 |
| No. 14 | 3 | 4 | 1 | 3 | 5 | 2 |
| No. 15 | 3 | 5 | 2 | 4 | 1 | 3 |
| No. 16 | 4 | 1 | 4 | 2 | 5 | 3 |
| No. 17 | 4 | 2 | 5 | 3 | 1 | 4 |
| No. 18 | 4 | 3 | 1 | 4 | 2 | 5 |
| No. 19 | 4 | 4 | 2 | 5 | 3 | 1 |
| No. 20 | 4 | 5 | 3 | 1 | 4 | 2 |
| No. 21 | 5 | 1 | 5 | 4 | 3 | 2 |
| No. 22 | 5 | 2 | 1 | 5 | 4 | 3 |
| No. 23 | 5 | 3 | 2 | 1 | 5 | 4 |
| No. 24 | 5 | 4 | 3 | 2 | 1 | 5 |
| No. 25 | 5 | 5 | 4 | 3 | 2 | 1 |

### 2.2. Indexes Determination

Gas–liquid contact with the co-current flow can be found in Figure 3a, the simulated flue gas, $CO_2$(A) + $N_2$(B), go through the distributor forming small bubbles and mixing with absorbent in the bubble-column scrubber. The $CO_2$ gas from the gas film diffuses into the liquid film and is absorbed by the liquid as shown in Figure 3b. At the interface, the $CO_2$ gas follows Henry's law and the absorption rate accompanying with two-film model becomes:

$$-r_A = (K_G a)_{loc}(C_A - HC_{LA}).\qquad(15)$$

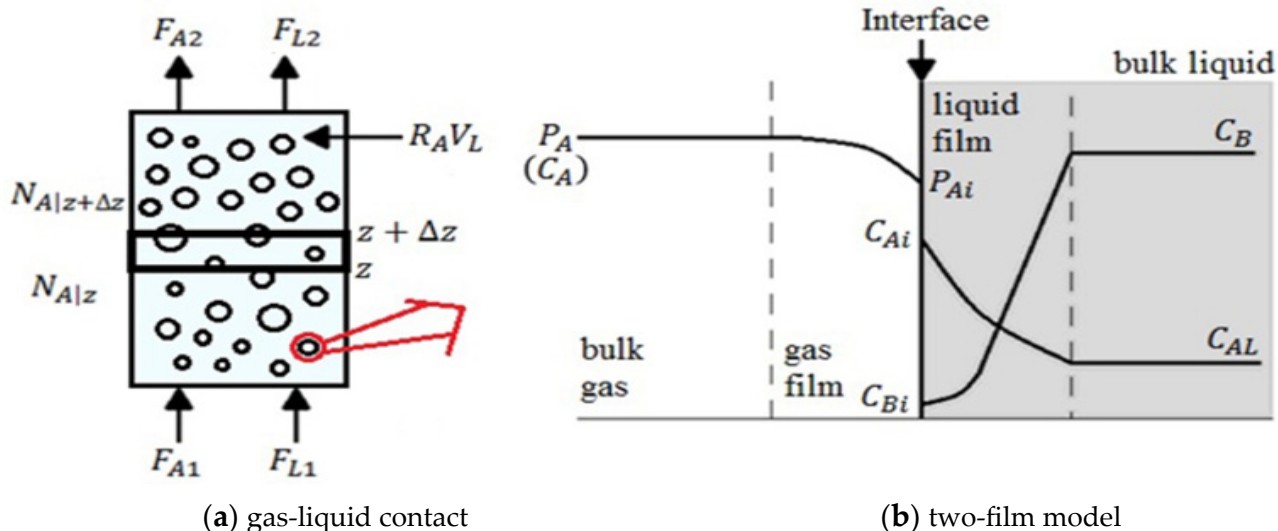

**(a)** gas-liquid contact          **(b)** two-film model

**Figure 3.** Absorption of $CO_2$ in the scrubber accompanied with two-film model.

In order to obtain the absorption rate and overall mass-transfer coefficient, the isothermal mass balance at a steady-state condition is used:

(Mass transfer of $CO_2$ at inlet) $-$ (Mass transfer of $CO_2$ at outlet) = Absorption rate of $CO_2$. (16)

The equations can be expressed below:

$$(N_A|_z - N_A|_{z+\Delta z})A = (-r_A)(A\Delta z)\varepsilon_L \tag{17}$$

where $N_A$ is the molar flux of component A. Alternatively, Equation (17) becomes:

$$(F_A|_z - F_A|_{z+\Delta z}) = (-r_A)(A\Delta z)\varepsilon_L \tag{18}$$

where $N_A A = F_A$, the molar flow rate. Take limit, Equation (18) becomes

$$-\frac{dF_A}{dz} = (-r_A)(A\varepsilon_L). \tag{19}$$

Integrate the Equation (19), it is

$$-\int_{F_{A1}}^{F_{A2}} dF_A = \int_0^L A\varepsilon_L dz = \int_0^{V_L} (-r_A)dV_L, \tag{20}$$

$$F_{A1} - F_{A2} = R_A V_L, \tag{21}$$

or

$$R_A = \frac{F_{A1} - F_{A2}}{V_L}. \tag{22}$$

Equation (22) shows that the overall absorption rate can be calculated from the measured input and output $CO_2$ gas molar rates and liquid volume. Because the molar flow rate of inert gas ($N_2$) is equal to $F_{A1}(1 - y_{A1})/y_{A1}$, the molar flow rate of carbon dioxide at the outlet ($F_{A2}$) is equal to $F_{A1}[(1 - y_{A1})/y_{A1}][y_{A2}/(1 - y_{A2})]$. Therefore, Equation (22) can be rewritten as:

$$R_A = \frac{F_{A1}}{V_L}[1 - (\frac{1 - y_{A1}}{y_{A1}})(\frac{y_{A2}}{1 - y_{A2}})]. \tag{23}$$

Equation (23) shows that absorption rate can be evaluated when $F_{A1}$, $V_L$, $y_{A1}$, and $y_{A2}$ are given. In addition, Equation (17) can be written as

$$(u_z C_A|_z - u_z C_A|_{z+\Delta z})A = (-r_A)(A\Delta z)\varepsilon_L \tag{24}$$

where $u_z$ is the linear velocity of gas through the column. Substitute Equation (15) into Equation (24) and it is divided by $\Delta z$ and take the limit, the equation becomes

$$-u_z \frac{dC_A}{dz} = (-r_A)\varepsilon_L = (K_G a)_{loc}(C_A - HC_{LA})\varepsilon_L. \tag{25}$$

For most systems, $C_A >> HC_{LA}$ [49]. Therefore, Equation (25) can be rearranged and integration, which is

$$-u_z \int_{C_{A1}}^{C_{A2}} \frac{dC_A}{C_A} = \int_0^L (K_G a)_{loc} dz. \tag{26}$$

Finally, the overall mass-transfer coefficient becomes

$$K_G a(s^{-1}) = \frac{u_z}{\varepsilon_L L} \ln\frac{C_{A1}}{C_{A2}} = \frac{Q_g(L/s)}{V_L(L)} \ln\frac{C_{A1}(mol/L)}{C_{A2}(mol/L)}, \tag{27}$$

where $K_G a$ is defined as the following:

$$K_G a = \frac{1}{L}\int_0^L (K_G a)_{loc} dz. \tag{28}$$

Equation (27) shows that overall mass-transfer coefficient can be estimated when $C_{A1}$, $C_{A2}$, $V_L$, and $Q_g$ are measured. In addition, the absorption efficiency and scrubbing factor are defined below:

$$E_F(\%) = (\frac{y_{A1} - y_{A2}}{y_{A1}}) \times 100\%, \tag{29}$$

$$\varphi(mol\,mol^{-1}L^{-1}) = \frac{F_G(mol/s)E_F(\%)(10^{-2})}{V_b(L)F_L(mol/s)}. \tag{30}$$

Heat duty of solvent regeneration includes three parts, i.e., heat of adsorption ($q_{ads}$), sensitive heat ($q_{sen}$), and heat of evaporation ($q_{sol}$):

$$\begin{aligned} q(G\quad Jt^{-1}) &= q_{ads}(GJ/t) + q_{sen}(GJ/t) + q_{sol}(GJ/t) \\ &= \Delta H^{ad}(GJ/t) + \frac{m_{sol}(kg)C_p(kJkg^{-1}K^{-1})\Delta T(K)}{\Delta m_{CO_2}(kg)} + \frac{\Delta m_1(kg)}{\Delta m_{CO_2}(kg)}\Delta H^{vap}(GJ/t) \end{aligned} \tag{31}$$

They can be determined when thermal data are available [32,50–54]. Thermal data used here include those of heat capacity [50], the heat of absorption [32,52–54], and latent heat [51]. In Equation (31), $C_p$ is the heat capacity of scrubbed solutions, $\Delta H^{ad}$ the heat of absorption, $\Delta T$ the temperature difference, $m_{sol}$ the mass of regeneration solution, $\Delta m_{CO_2}[kg]$ the mass loss of $CO_2$ after stripping, $\Delta H_{vap}$ the heat of evaporation, and $\Delta m_1$ the scrubbed solution loss during stripping.

## 3. Experimental Procedure

### 3.1. Absorption Test

The experimental devices are depicted in Figure 4, including the bubble column, tubing pumps for gas-flow and liquid-flow, a mass flow controller, pH-meter, $CO_2$ meter, gas heating chamber, and cooler. To start with, the desired mixed amine concentration was prepared using distilled water. Next, the flow rate of carbon dioxide and nitrogen was adjusted using a mass flow controller into the proportion of 15% of $CO_2$, maintaining the gas inlet temperature at the bottom of the scrubber at 50 °C. The mixed amine was placed into the scrubber after the desired temperature and $CO_2$ concentration were achieved, and the experiment was started. During the experimentation, the pH of the solution, liquid temperature, gas inlet temperature, gas outlet temperature, pressure, and $CO_2$ concentration were recorded every 5 min. The liquid at the outlet was also withdrawn for titration to observe the concentration of carbonate in the scrubbing solution. At the end of steady-state operation, the liquid input was closed, and the solution was withdrawn to measure the volume of liquid ($V_L$) in the scrubber using a tubing pump. Using the measured data, including $P$, $y_{A1}$, $y_{A2}$, T, and $V_L$, all the indicators can be evaluated.

### 3.2. Regeneration Test

The equipment for regeneration tests is similar to those used in the previous work [17]. First, 0.05 kg scrubbed solution was prepared, and the ball condenser tube, three-neck round flask, heating system, and cooling circulator were assembled. The input cooling water temperature was set to 5 °C. When the heating oil temperature reached the desired value (100, 110, or 120 °C) and the cooling circulator temperature was stable, the 0.05 kg scrubbed solution was poured into the flask, and the magnetic stirrer was switched on. The experimental time was at least 60 min, and the temperature change was recorded once every 5 min. When the experiment was finished, the heating controller and cooling circulator were switched off, the mass of the scrubbed solution was measured, and the samples were taken for titration to measure the $CO_2$ loading.

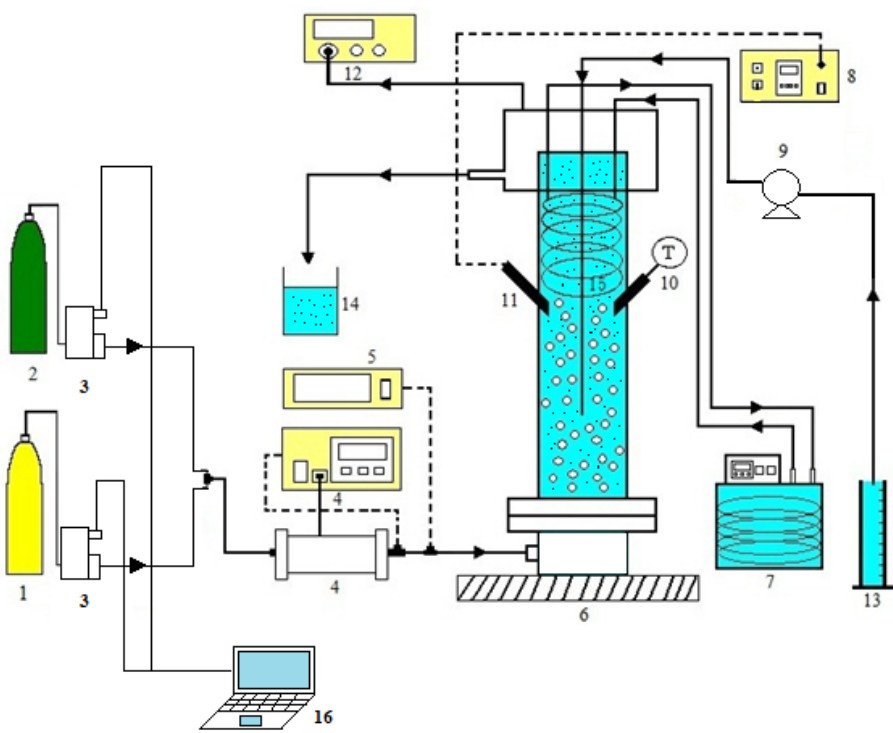

1. N₂-gas tank
2. CO₂-gas tank
3. Mass-flow controller
4. Gas heating system
5. Pressure gauge
6. Bubble-column scrubber
7. Cooling system
8. pH-meter
9. Tubing pump
10. Thermometer
11. pH-electrode
12. CO₂-meter
13. Amines vessel
14. Reservoir
15. Heating coil
16. Notebook computer

**Figure 4.** Experimental device for $CO_2$ capture test.

## 4. Results and Discussions

### 4.1. Steady-State Operation

A plot of Y versus t for No. 1 showing the variation in measured data with time is shown in Figure 5, wherein Y is defined as the ratio of the measured value to the initial value or the setting value. The figure shows that $CO_2$ concentration, pH, liquid temperature, gas temperature at the inlet, and gas temperature at the top of the scrubber were maintained constant after 30 min. This variation showed that the system reached a steady-state condition. Due to this, the measured data could be used to evaluate the values of indexes. Calculated data for $E_F$, $R_A$, $K_G a$, and $\phi$ are listed in Table 4. The range of data was 56.58–100.0%, $4.42 \times 10^{-4}$–$18.95 \times 10^{-4}$ mol s$^{-1}$ L$^{-1}$, 0.1195–0.9139 s$^{-1}$, and 0.0433–0.2923 mol-$CO_2$ mol-solvent$^{-1}$ L$^{-1}$ for $E_F$, $R_A$, $K_G a$, and $\phi$, respectively. The steady-state $pH$ values were 9.60–11.42, depending on the operating conditions. In addition, $\gamma$-values were in the range of 0.2942–2.3056. All data were analyzed to search further for the optimum mixed amine.

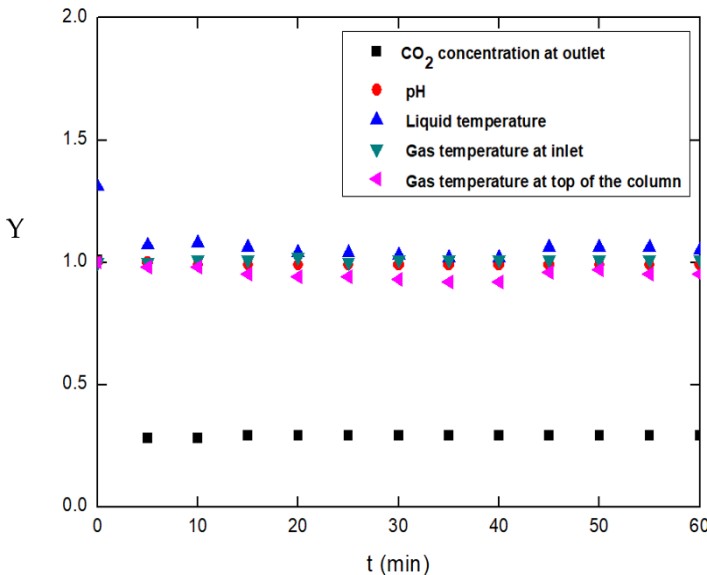

**Figure 5.** Variation of measurements showing steady-state condition (No. 1).

**Table 4.** Values of indexes for different runs.

| No. | $pH$ | $E_F$ (%) | $R_A(10^4)$ (mol/L·s) | $K_G a$ (1/s) | $\gamma$ (–) | $\phi$ (mol-$CO_2$/L·mol-Solvent) |
|---|---|---|---|---|---|---|
| 1 | 10.18 | 71.05 | 4.42 | 0.1195 | 1.2147 | 0.1485 |
| 2 | 10.75 | 80.26 | 7.74 | 0.2543 | 0.8856 | 0.1223 |
| 3 | 10.70 | 84.00 | 10.52 | 0.3808 | 0.6943 | 0.0990 |
| 4 | 10.56 | 84.42 | 14.68 | 0.5241 | 0.5902 | 0.0868 |
| 5 | 10.96 | 88.00 | 18.95 | 0.7740 | 0.5055 | 0.0755 |
| 6 | 10.55 | 90.67 | 13.54 | 0.6006 | 0.7201 | 0.1108 |
| 7 | 11.42 | 80.00 | 12.93 | 0.4179 | 0.5831 | 0.0792 |
| 8 | 10.20 | 58.67 | 12.46 | 0.2875 | 1.7294 | 0.1722 |
| 9 | 11.35 | 94.67 | 5.47 | 0.2825 | 0.3428 | 0.0551 |
| 10 | 10.65 | 89.33 | 8.21 | 0.3424 | 0.9110 | 0.1382 |
| 11 | 10.16 | 69.33 | 14.27 | 0.3962 | 1.3963 | 0.1643 |
| 12 | 10.61 | 94.67 | 5.79 | 0.3007 | 0.3086 | 0.0496 |
| 13 | 11.02 | 86.67 | 7.81 | 0.2980 | 0.2942 | 0.0433 |
| 14 | 9.60 | 79.73 | 9.98 | 0.3285 | 0.7979 | 0.1066 |
| 15 | 10.09 | 56.58 | 10.05 | 0.2283 | 2.2523 | 0.2192 |
| 16 | 10.90 | 90.79 | 8.30 | 0.3534 | 0.3055 | 0.0477 |
| 17 | 10.11 | 74.03 | 10.10 | 0.2938 | 1.0995 | 0.1419 |
| 18 | 9.85 | 72.73 | 12.79 | 0.3666 | 2.3056 | 0.2923 |
| 19 | 10.64 | 68.00 | 15.76 | 0.4150 | 1.3803 | 0.1593 |
| 20 | 10.98 | 96.00 | 5.60 | 0.3073 | 0.3149 | 0.0513 |
| 21 | 10.57 | 81.33 | 13.38 | 0.4497 | 0.6659 | 0.0919 |
| 22 | 10.43 | 77.33 | 15.62 | 0.4884 | 1.7929 | 0.2354 |
| 23 | 11.03 | 100.00 | 5.43 | 0.9139 | 0.3323 | 0.0564 |
| 24 | 9.96 | 89.33 | 7.71 | 0.3326 | 1.0898 | 0.1653 |
| 25 | 10.56 | 88.00 | 10.77 | 0.4339 | 0.8702 | 0.1300 |

### 4.2. Effects of Mixed Amines on the Indexes

Factor A represents mixed amine, as shown in Table 3, including A1 (MEA; Nos. 1–5), A2 (MEA + DIPA; Nos. 6–10), A3 (MEA + TEA; Nos. 11–15), A4 (MEA + AMP; Nos. 16–20), and A5(MEA + PZ; Nos. 21–25). The combination of operating conditions for B, C, D, E, and F are all involved in A1-A5. Considering $E_F$ as an example, the mean values for A1, A2, A3, A4, and A5 were 81.55%, 82.67%, 77.40%, 80.31%, and 87.20%, respectively. The sequence of A was found to be A5 > A2 > A1 > A4 > A3, showing that mixed amine (MEA + PZ) for $E_F$ was the best. The result also showed that the addition of

PZ into MEA could enhance the absorption efficiency of $CO_2$. Likewise, the mean values for other indexes are listed in Table 5. The importance sequence for $R_A$ was in the order of A1 > A5 > A2 > A4 > A3, meaning that the rate of absorption of MEA (A1) was the highest. In addition, the importance sequence for $K_Ga$ in the order of A5 > A1 > A2 > A4 > A3, showing (MEA + PZ) could promote the overall mass transfer coefficient up to 28% as compared with MEA (A1), while the results of other mixed amines were contrary. On average, compared with MEA, the use of MEA + PZ mixed solvent could reduce the scrubber size by about 30%. Finally, the importance sequence for $\phi$ was in the order of A4 > A5 > A3 > A2 > A1, showing that the addition of DIPA, TEA, AMP, and PZ into MEA could improve the $CO_2$ loading, while MEA + AMP (A4) increased by about 30%. The score for each mixed amine needs to be defined for quantitative comparison. According to the sequence of importance, the corresponding points were 5, 4, 3, 2, and 1. Using $E_F$ as an example (A5 > A2 > A1 > A4 > A3), the points for A1, A2, A3, A4, and A5 were 3, 4, 1, 2, and 5, respectively. All the scores and mean values are listed in Table 6. The whole sequence of importance was in the order of A5(4.5) > A1(3.25) > A2(3) > A4(2.75) > A3(1.5). The analysis shows that MEA + PZ (A5) was the best-mixed amine.

**Table 5.** Importance sequence analysis for various indicators.

| Indexes | A1 | A2 | A3 | A4 | A5 |
|---|---|---|---|---|---|
| $E_F$ | 81.54 | 82.67 | 77.40 | 80.31 | 87.20 |
| $R_A$ $(10^4)$ | 11.26 | 10.52 | 9.58 | 10.51 | 10.58 |
| $K_Ga$ | 0.410 | 0.386 | 0.310 | 0.347 | 0.524 |
| $\phi$ | 0.1064 | 0.1111 | 0.1166 | 0.1385 | 0.1358 |

**Table 6.** Points for various indicators.

| Indexes | A1 | A2 | A3 | A4 | A5 |
|---|---|---|---|---|---|
| $E_F$ | 3 | 4 | 1 | 2 | 5 |
| $R_A$ | 5 | 3 | 1 | 2 | 4 |
| $K_Ga$ | 4 | 3 | 1 | 2 | 5 |
| $\phi$ | 1 | 2 | 3 | 5 | 4 |
| mean | 3.25 | 3 | 1.5 | 2.75 | 4.5 |

*4.3. Taguchi Analysis*

Using $E_F$ as an example, the S/N (signal/noise) ratio analysis for larger-the-better can be determined by Equation (32):

$$\frac{S}{N} = -10 \times \log\left(\frac{1}{n}\sum_1^n z_i^2\right), \tag{32}$$

where n is the number of data and $z_i$ is the outcome data such as $E_F$, $R_A$, $K_Ga$, and $\phi$. Using No. 1 as an example, for A1 mixed solvent, the data for $E_F$ presented in Table 5 were 71.05%, 80.26%, 84.00%, 84.42%, and 88.00%, respectively. On substituting these values into Equation (32), A1 was estimated to be 38.1559. In the same method, all S/N ratios were evaluated and are listed in Table 7. The bracket words shown in this table are the maximum values for each factor, which includes the optimum condition, A5B4C5D1E5F5. The 'DELTA' value in this table is the difference between the maximum and minimum values of each factor, such as (A5–A3), yielding 1.37. According to the reported data, the parameter sequence was found to be in the order of E > D > A > F > C > B. Using the same analysis, the optimum condition and parameter sequence for other indexes could be determined and are listed in Table 8, in which the mixed amines in optimum conditions were A5, A2, A5, and A1, respectively. This supports the finding presented in Section 3.2 that MEA + PZ is the best amine.

**Table 7.** S/N ratio analysis for $E_F$ giving optimum condition and parameter significance.

| Level | A | B | C | D | E | F |
|---|---|---|---|---|---|---|
| 1 | 38.1559 | 37.9573 | 37.7625 | (38.9978) | 36.5413 | 37.784 |
| 2 | 37.9301 | 38.1093 | 37.4206 | 38.7922 | 37.9967 | 37.6308 |
| 3 | 37.3404 | 37.6729 | 37.7078 | 38.343 | 38.262 | 37.674 |
| 4 | 37.8643 | (38.2346) | 37.9977 | 37.2163 | 38.7237 | 38.2035 |
| 5 | (38.7104) | 37.9347 | (38.4925) | 36.9414 | (38.7678) | (38.6857) |
| DELTA | 1.37 | 0.5617 | 1.0719 | 2.0564 | 2.2265 | 1.0549 |
| RANK | 3 | 6 | 5 | 2 | 1 | 4 |

**Table 8.** Optimum condition and parameter sequence for all indicators.

| Indicators | Optimum Condition | Parameter Sequence |
|---|---|---|
| $E_F$ (No. 26) | A5B4C5D1E5F5 | E > D > A > F > C > B |
| $R_A$ (No. 27) | A2B2C3D5E4F5 | D > C > E > F > A > B |
| $K_G a$ (No. 28) | A5B3C3D5E5F4 | E > A > D > F > C > B |
| $\phi$ (No. 29) | A1B4C1D5E1F4 | E > C > D > A > B > F |

Based on points (0–5) analysis for six factors, the importance of parameters among the whole factors became E (4.5) > D (3.75) > C (2.5) = A (2.5) > F (1.5) > B (0.25), showing that the concentration of mixed amine (E) is the most important factor obtained. However, the optimum condition needs to be verified further.

*4.4. Verifications of Optimum Conditions*

The experiments were performed further according to optimum conditions listed in Table 8 and the results are listed in Table 9. The reported values were 100%, $19.9584 \times 10^{-4}$ mol/s·L, 1.1675 1/s, and 0.4148 mol-CO$_2$ /L·mol-solvent for $E_F$, $R_A$, $K_G a$, and $\phi$, respectively. The values are listed in bracket words. Compared with Nos. 1–25, the obtained values were all maximum, i.e., the results could be confirmed. This also demonstrates the feasibility of the Taguchi experimental design in this study.

**Table 9.** Verifications of optimum conditions.

| No. | $E_F$ (%) | $R_A \times 10^4$ (mol/s·L) | $K_G a$ (1/s) | $\phi$ (mol-CO$_2$/L·mol-Solvent) |
|---|---|---|---|---|
| 1–25 | 56.58–100.0 | 4.4153–18.9500 | 0.1195–0.9139 | 0.0433–0.2923 |
| $E_F$(26) | (100.0) | 5.6630 | 1.1698 | 0.0313 |
| $R_A$(27) | 90.67 | (19.9584) | 0.8784 | 0.1507 |
| $K_G a$(28) | 96.67 | 19.2220 | (1.1675) | 0.1316 |
| $\phi$(29) | 82.67 | 17.3165 | 0.6144 | (0.4148) |

*4.5. Effects of Variables on the Indexes*

Figure 6 presents a plot of $E_F$ versus $\gamma$ for different mixed amines. According to the trend, $E_F$ decreases with an increase in $\gamma$, while mixed amine A5 (MEA + PZ) yielded a higher $E_F$ at the same $\gamma$, suggesting that PZ can promote mass transfer and, hence, the absorption efficiency. There are two points that affect the $\gamma$ and hence $E_F$ value; one is the resident time of CO$_2$ gas and the other is the solvent molar flow rate. A lower residence time meaning a higher gas flow rate leads to reduce $E_F$, while the $E_F$ can be increased when a higher molar flow rate was used. Nonetheless, the $E_F$ is also affected by temperature and pH, which are not included in Figure 6a. Therefore, a regression was required to assess the influences of $\gamma$, $T$, and pH on the $E_F$. The results are shown below:

$$E_F(\%) = 3.0532 \times 10^2 \exp\left(-\frac{4137}{R(\text{JK}^{-1}\text{mol}^{-1})\,T(\text{K})}\right)\gamma^{-0.1801} pH^{0.09684}. \tag{33}$$

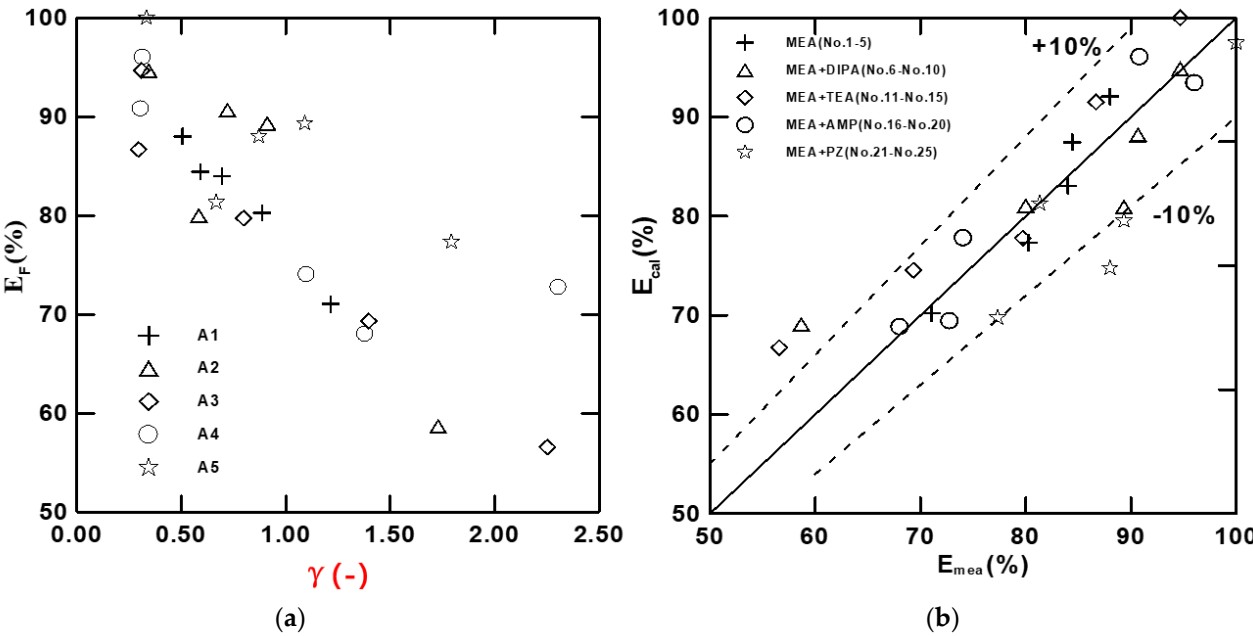

**Figure 6.** Effect of parameter on $E_F$: (**a**) A plot of $E_F$ versus $\gamma$ for different mixed amines. (**b**) Confidence of Equation (33).

The $R^2$ was found to be 0.7453 (>0.5), indicating that the powder law model is suitable for $E_F$ and that the results are reliable. $E_F$ increased with an increase in t and pH and decreased with an increase in $\gamma$. Figure 6b is a plot of calculated values versus measured values; almost all data were within 10% error, showing the confidence of Equation (33).

A similar regression was also performed for $R_A$ and $K_Ga$, but the R squares for both were <0.5 showing the differences in solvents. Due to this, the regressions could be carried out for different solvents and were expressed as: $\xi = \alpha pH^a \gamma^b t^c$. Tables 10 and 11 show both parameters and $R^2$ for different solvents. A2 and A3 were excluded because $R^2 < 0.5$, and $R_A$ decreased with an increase in $\gamma$ for A1 and decreased with an increase in t for A5. In addition, $R_A$ increased with an increase in $pH$, $\gamma$, and $t$, respectively. According to Figure 7a, a plot of calculated data versus measured data showed that the error limit for most data was within 20%. A2 and A3 showed some scattering data; conversely, for A1, $K_Ga$ decreased with an increase in $\gamma$ and $K_Ga$ and increased with an increase in pH, $\gamma$, and t, respectively. Figure 7b shows that a plot of calculated data versus measured data showed that the errors of most data were within 20%. Some scattering data were also observed for A2 and A3. The results also reveal that the effect of terminal pH was more significant for $R_A$ and $K_Ga$, while that of was $\gamma$ minor.

**Table 10.** Regression parameters of $R_A$ for different solvents.

| Mixed Solvent | $\alpha$ | a | b | c | $R^2$ |
|---|---|---|---|---|---|
| A1 | $1.067 \times 10^{-6}$ | 0.9495 | $-0.7488$ | 1.2368 | 0.9966 |
| A2 | $2.459 \times 10^{-7}$ | 3.4971 | 0.6338 | 0.03703 | <0.5 |
| A3 | $6.959 \times 10^{-3}$ | $-0.3646$ | 0.2882 | $-0.3088$ | <0.5 |
| A4 | $5.063 \times 10^{-21}$ | 15.3879 | 0.9809 | 1.0931 | 0.6863 |
| A5 | $3.608 \times 10^{-5}$ | 2.6498 | 0.5845 | $-0.7938$ | 0.7099 |

**Table 11.** Regression parameters of $K_G a$ for different solvents.

| Mixed Solvent | $\alpha$ | a | b | c | $R^2$ |
|---|---|---|---|---|---|
| A1 | $3.286 \times 10^{-6}$ | 3.0754 | −1.1424 | 1.1148 | 0.9994 |
| A2 | $8.256 \times 10^{-12}$ | 8.4837 | 0.5876 | 1.2734 | <0.5 |
| A3 | 5.733 | −1.5210 | −0.1029 | 0.1648 | <0.5 |
| A4 | $1.576 \times 10^{-20}$ | 16.8625 | 0.7071 | 1.4307 | 0.9680 |
| A5 | $4.146 \times 10^{-14}$ | 11.7191 | 0.08486 | 0.7232 | 0.9792 |

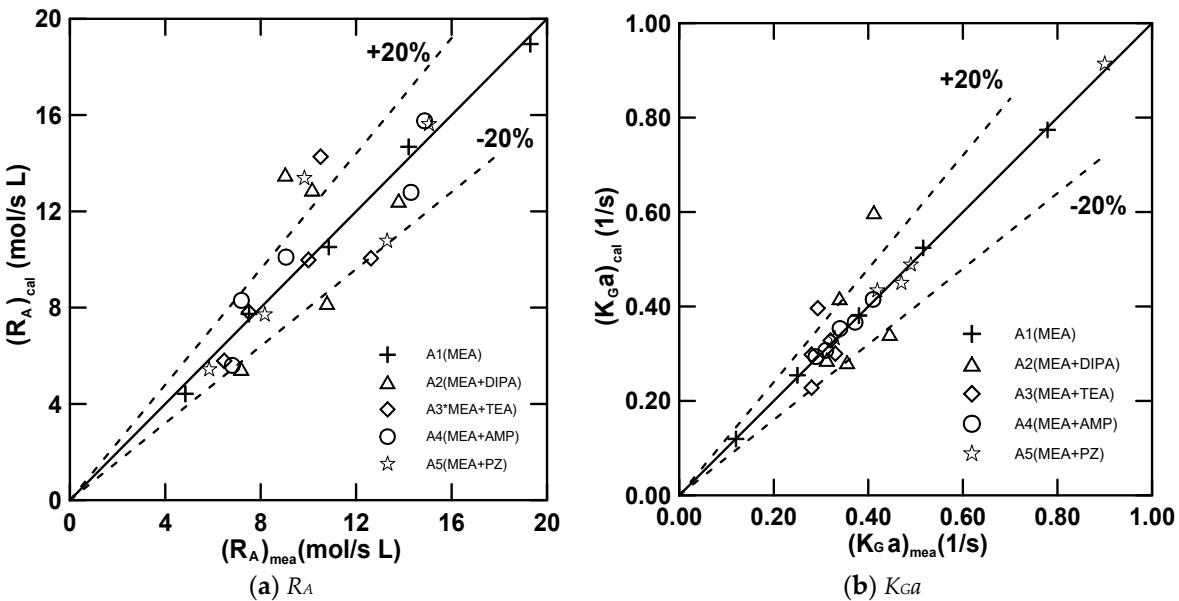

(**a**) $R_A$       (**b**) $K_G a$

**Figure 7.** A plot of calculated value versus measured values for $R_A$ and $K_G a$.

*4.6. Solvent Regeneration Test*

Solvent regeneration tests for four optimum conditions were explored further. The scrubbed solution included two MEA + PZ mixed amines (No. 26 and No. 28), one MEA + DIPA (No. 27) scrubbed solution, and one MEA (No. 29) scrubbed solution. The tested temperatures were set to 100, 110, and 120 °C, separately. The overall heat of regeneration was 3.39–8.45 GJ/t, within the range of 2 GJ/t and 12 GJ/t reported in the literature [13,55,56]. Figure 8a presents a plot of q versus t showing the effect of different solvents on the heat of solvent regenerations. The plots showed that the minimum energy requirement was at 110 °C for mixed solvents, No. 26-No. 28., while the heat of regeneration decreased with a decrease in temperature for No. 29 (MEA single solvent). Comparing No. 26 and No. 28 (both MEA + PZ), at the same factor E (3M) but for factor B, were 20% and 15%, respectively. The final loadings were 0.0866 and 0.1349 mol-$CO_2$/mol-solvent for No. 26 and No. 28, respectively. The heat of regeneration for No. 28 was lower than that for No. 26, suggesting a lower heat of regeneration at higher loading than that at lower loading. A similar result was reported in the literature [13]. The sequence of the heat of regeneration was in the order of MEA > MEA + PZ > MEA + DIPA (Figure 6).

The results of individual energy estimation revealed that $q_{sen}$ were in the range 0.45–1.43 GJ/t, $q_{sol}$ were in the range 1.04–5.51 GJ/t, and $q_{ads}$ were in the range of 1.83–1.93 GJ/t. The fractions ($F$) of individual required energies were in the range of 0.102–0.232, 0.317–0.652, and 0.229–0.577 for $q_{sen}$, $q_{sol}$, and $q_{ads}$, respectively. For further discussion, the distribution in individual energy was plotted and is shown in Figure 8b, wherein $q_{sen}$ and $q_{sen}$ both increase with an increase in $F$, while $q_{ads}$ causes a small change in $F$ in the range of 0.229–0.577. When $F > 0.4$, the energy required was in the order of $q_{sol} > q_{ads} > q_{sen}$. Three major choices were found to decrease q: one is selecting a lower heat of absorption solvent; second, using a higher concentration solvent to reduce the

heat of evaporation; third, an increase in $\gamma$, which can increase $CO_2$ loading and hence decreasing q.

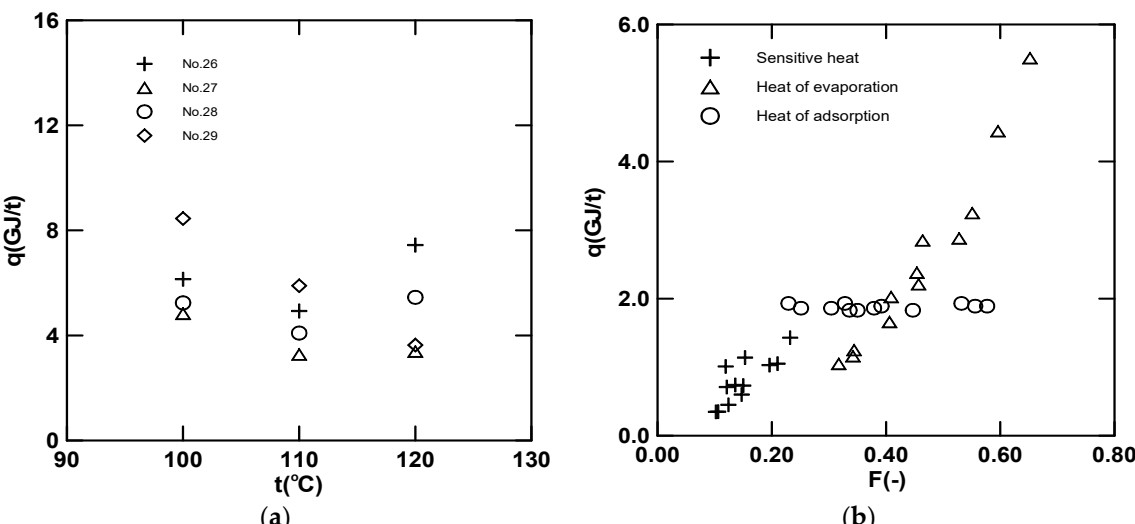

**Figure 8.** Heat of regeneration and mechanism. (**a**) A plot of q versus t at different scrubbed solutions. (**b**) A plot of q versus *F* showing individual required energy distribution.

## 5. Conclusions

Using the Taguchi experimental design, the experimental number of runs could be reduced from 15,625 to 25, cutting the experimental cost up to 99.84%. A continuous bubble-column scrubber was successfully used to assess the performance of mixed solvents. Solvent regeneration tests were also carried out. It is a shortcut and also a quick method to obtain the best solvent, optimum condition, and the order of parameter importance. Based on the six factors analysis here, the importance of parameters of the whole factors was found to be in the order of E > D > C = A > F > B, showing the concentration of mixed amine (E) is the most significant factor, while the ratio of mixed amine is a minor one. The priority sequence of mixed amine was found to be in the range of A5 > A1 > A2 > A4 > A3, and the regeneration energy was in the range of A1 > A5 > A2, showing that A5 (MEA + PZ) is the best solvent. The absorption efficiency could be controlled to a higher value (>90%) when $\gamma < 0.5$. The sequence of promoting the $CO_2$ loading was AMP > PZ > TEA > DIPA. According to overall mass-transfer coefficients, the scrubber size could be reduced by 30% when using A5 (MEA + PZ) mixed amine as compared with A1 (MEA) solvent. Alternatively, the indexes were expressed in terms of pH, $\gamma$, and t with different parameters, depending on the mixed solvent used. The better conditions were found to be B = 15–20%, C = 250–350 mL/min, D = 4–12 L/min, and E = 3 M; F = 40–45 °C. In addition, the minimum heat of regeneration was achieved when the operating heating temperature was 110 °C. The individual energy required was in the sequence of $q_{sol} > q_{ads} > q_{sen}$ when $F > 0.4$. The heat of regeneration of scrubbed solutions could be controlled under 3 GJ/t or lower when increasing the concentration of mixed amines or using a solvent with a lower heat of absorption.

**Author Contributions:** Conceptualization, P.-C.C.; methodology, P.-C.C.; software, P.-C.C. and J.-H.J.; validation, T.-W.W., C.-Y.Y., K.-Y.W. and C.-M.C.; formal analysis, T.-W.W., C.-Y.Y., K.-Y.W. and C.-M.C.; investigation, J.-H.J., T.-W.W., C.-Y.Y., K.-Y.W. and C.-M.C.; resources, P.-C.C.; data curation, P.-C.C.; writing—original draft preparation, P.-C.C., T.-W.W., C.-Y.Y., K.-Y.W. and C.-M.C. All authors have read and agreed to the published version of the manuscript.

**Funding:** This research was funded by MOST, Taiwan, grant number 110-2221-E-262-003.

**Institutional Review Board Statement:** Not applicable.

**Informed Consent Statement:** We understand the general purposes, risks and methods of this research. We consent to participate in the research project and the following has been explained to me: the research may not be of direct benefit to us.

**Data Availability Statement:** Data are available on request only due to ethical, legal or commercial reasons.

**Conflicts of Interest:** The authors declare no conflict of interest.

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
