# Peer review of "Capture of CO2 Using Mixed Amines and Solvent Regeneration in a Lab-Scale Continuous Bubble-Column Scrubber"

_applsci, doi:10.3390/app13127321_

Round 1

Reviewer 1 Report

The manuscript reports CO2 capture results with five amine solvents in a bubble column. There are following comments for modifications.

1. Usually such kind of CO2 absorption should be carried out in packed columns with a gas continuous phase and a liquid distributed phase, which lead low energy-consumption for solvent degeneration. If bubble column is used, a large liquid rate should be used which lead to high energy consumption. It needs more and detailed discussion about why a bubble column is used here.

2. The data measured and the calculated data as KG cannot be used in modelling for packed columns.

Author Response

Please attached file. Thank you

Reviewer 2 Report

Comments

The manuscript reports the carbon capture with mixed amines in a lab-sclae continuous bubble-column scrubber. The factors including mixed amine, the ratio of mixed amines, the liquid feed flow, the gas flow rate, the concentration of mixed amines, and the liquid temperature were studied using a Taguchi experimental design. Indicators including absorption efficiency, absorption rate, overall mass-transfer coefficient, and absorption factor were estimated for searching the best conditions. The Taguchi experimental design showed a high feasibility in this study by saving much time and cost. However, the manuscript spends lots of space for explaining the experienal design and related discussion. The reason and possible mechanism for the optimum conditions were rarely dicussed. It is suggested to reduce the discussion of experimental design and add a Section 4.7 to explain the optimum condition. Some of the information in the introduction part may be linked with the discussion in 4.7 as the experimental results were the strong evidence to support the series of hypothesis and assumptions. All in all, the manuscript provides some useful information of absorbent and absorption process design for carbon capture from an engineering prospective. It may be worth of publication on Applied Sciences by addressing above mentioned issues.

In addition, please carefuly check the whole manuscript, examples of typos and style issues were listed as follow:

Line 19 and other parts of the manuscript.

·      CO2 should be CO2

Line 40

·      Please check the font of “grown”

Table 1

·      Please check the font of the symbol “ ºC ” in table.

Eqs.1

·      Please check the equation carefully.

Reviewer 3 Report

General comment: The article “Capture of CO2 Using Mixed Amines and Solvent Regeneration in A Lab-scale Continuous Bubble-column Scrubber” is a great effort by the authors. They have experimented almost all necessary directions. However, few comments are made to improve the quality of the manuscript.

Specific comments:

Comment 1: Line 29, 30: Please include the citation.

Comment 2:  please check the subtitle 4.1. Is it steady state?

Comment 3: Please check font sizes of the equations. It seems some equations are abruptly large. Try to make it uniform.

Comment 4: In the images also please make the font in the titles uniform. Example, Fig 4a the x-axis title is not bold but y-axis title is bold. 

Comment 5: In section 4.5 authors have explained what is there in the graphs but they should elaborate why it is happening. Because anyone can understand the increasing decreasing trend but the explanation for the trend is important.

Comment 6: If possible include a table of comparison with the existing data and new data. This will help the readers to understand the novelty of the work

Reviewer 4 Report

Manuscript ID: applsci-2421114

Title: Capture of CO2 Using Mixed Amines and Solvent Regeneration in A Lab-scale Continuous Bubble-column Scrubber
Authors: Pao Chi Chen*, J.H. Jhuang, T.W. Wu, C.Y. Yang, K.Y. Wang and C.M. Chen

This manuscript deals with the optimization of an amine-based CO2 scrubber column. The authors have explored the effect of monoethanolamine (MEA) being mixed with secondary amines (DIPA), tertiary amines, stereo amines, and piperazine (PZ) at diferent temperatuures and flow rates to the absorption efficiency (EF), absorption rate (RA), overall mass-transfer coefficient (KGa), and absorption factor (Ï•). They have used the Taguchi method to design their experiments for reducing the amount of workload. Overall, the work is well done and the authors have achieved valuable results.

The manuscript is structured in a way that makes sense and the authors have discussed most of the issues with storing CO2 in activated carbon, but there remain some parts that should be improved before the manuscript is publishable in Journal of CO2 Utilization.

Comments to the Authors:

1)      The main issue with the Taguchi method is that the parameters optimized cannot be interdependent. Here, the authors have selected the amine mixture type, the ratio of amines between each other, liquid flow rate, gas flow rate, the concentration of mixed amines, and the liquid temperature as the parameters. With their instrument design, the gas flow rate most certainly will have an effect on temperature as the gas bubbles flow from the bottom and the heating coils are at the top of the instrument. In addition, the amine mix type and rate will also likely affect this as they have different heat conductivity, and in turn, the liquid flow rate will also affect the mixing of the solution. How do the authors justify using the Taguchi experimental design in this case where the tested parameters are clearly dependent on each other?

2)      The authors have written on lines 19-20 that „The optimum conditions were validated and found to be 100 %, 19.96×10-4 mole/s·L, 1.2312 1/s, and 0.6891 mol-CO2/L·mol-solvent for EF, RA, KGa, and Ï•, respectively.“ These are results, not conditions. This sentence should be rewritten.

3)      The manuscript is written in a somewhat confusing way in parts. It should definitely be edited by a native speaker.

4)      Other, competing methods for absorption and utilization of CO2, such as the capture of CO2 in molten salts (W. Weng et al., J. Energy Chem. 28 (2019) 128–143; A.L. Remmel et al., ACS Sustain. Chem. Eng. 10 (2022) 134–145; S. Ratso et al., Green Chem. 23 (2021) 4435–4445; R. Jiang,  et al., Curr. Opin. Electrochem. 17 (2019) 38–46) should at least be mentioned in the introduction.

Minor remarks:

1)      There are some formatting errors, such as the text flowing randomly on page 9, and the text size differing on line 370. These should be corrected before publication.

The manuscript is written in a somewhat confusing way in parts. It should definitely be edited by a native speaker.

Round 2

Reviewer 1 Report

The manuscript reports CO2 absorption data with five solvents in a bubble column. The results are interesting to readers.

The comments from reviewers have been considered in the revision. 

I suggest to publish the article.